# Offretite Zeolite Single Crystals Synthesized by Amphiphile-Templating Approach

**DOI:** 10.3390/molecules26082238

**Published:** 2021-04-13

**Authors:** Eng-Poh Ng, Nur Hidayahni Ahmad, Fitri Khoerunnisa, Svetlana Mintova, Tau Chuan Ling, T. Jean Daou

**Affiliations:** 1School of Chemical Sciences, Universiti Sains Malaysia, Penang 11800 USM, Malaysia; nurhidayahni@gmail.com; 2Department of Chemistry, Universitas Pendidikan Indonesia, Setiabudi 229, Bandung 40154, West Java, Indonesia; fitri@upi.edu; 3Laboratoire Catalyse et Spectrochimie (LCS), Normandie Université, ENSICAEN, UNICAEN, CNRS, 14050 Caen, France; mintova@ensicaen.fr; 4Institute of Biological Sciences, Faculty of Science, University of Malaya, Kuala Lumpur 50603, Malaysia; tcling@um.edu.my; 5Institut de Science des Matériaux de Mulhouse (IS2M), CNRS, Axe Matériaux à Porosité Contrôlée (MPC), Université de Haute-Alsace (UHA), 3 bis rue Alfred Werner, 68093 Mulhouse, France; 6Université de Strasbourg, 67000 Strasbourg, France

**Keywords:** hierarchical zeolites, crystal growth, offretite zeolite, amphiphile-templating approach, acylation of 2-methylfuran

## Abstract

Offretite zeolite synthesis in the presence of cetyltrimethylammonium bromide (CTABr) is reported. The offretite crystals were synthesized with a high crystallinity and hexagonal prismatic shape after only 72 h of hydrothermal treatment at 180 °C. The CTABr has dual-functions during the crystallization of offretite, viz. as structure-directing agent and as mesoporogen. The resulting offretite crystals, with a Si/Al ratio of 4.1, possess more acid sites than the conventional offretite due to their high crystallinity and hierarchical structure. The synthesized offretite is also more reactive than its conventional counterpart in the acylation of 2-methylfuran for biofuel production under non-microwave instant heating condition, giving 83.5% conversion with 100% selectivity to the desired product 2-acetyl-5-methylfuran. Hence, this amphiphile synthesis approach offers another cost-effective and alternative route for crystallizing zeolite materials that require expensive organic templates.

## 1. Introduction

Zeolites are crystalline microporous materials made up of tetrahedral TO_4_ (T = Si or Al) repeating units [1]. These porous solids are widely used in catalysis, separation, and ion exchange processes due to their unique structure with defined size and shape [2,3,4]. Currently, 253 types of zeolite structures have been recognized by the International Zeolite Association (IZA) [5]. The synthesis of zeolites can be performed by using various techniques such as organo-templating [6,7]; organotemplate-free [8,9]; ionothermal [10,11]; interconversion [12,13]; assembly, disassembly, organization, and reassembly (ADOR) [14,15]; and fluoride routes [16,17].

Offretite zeolite (OFF topology) has a 3-dimensional channel system containing 12-membered rings with a diameter 6.7 × 6.8 Å^2^. It is considered a very important zeolite due to its large pore size, which allows ease of molecular diffusion and accessibility [18]. To our knowledge, offretite can only crystallize in the presence of tetramethylammonium (TMA^+^) as a structure-directing agent (SDA) and both NaOH and KOH as mineralizers [19]. However, this zeolite seldom crystallizes completely free from erionite (ERI) and analcime (ANA) intergrowth, making the synthesis of pure OFF phase a challenge [20].

Recently, a new strategy for synthesizing offretite has been reported using new SDAs, such as pyrocatechol [21] and *p*-dioxane [22]. The resulting crystals possess different morphologies (oval, hexagonal, broccoli-like, prismatic) with merely microporosity due to the unique properties (e.g., size, shape, polarity, etc.) of the SDAs. Furthermore, offretite can also be readily crystallized via inter-zeolite conversion method using benzyltrimethylammonium hydroxide as an organic template, but the method is complicated and produces a small amount of erionite as a competing phase [23].

In the present work, we applied a combined rational strategy to synthesize large offretite single crystals with hierarchical porosity using a cationic amphiphile template (cetyltrimethylammonium bromide, CTABr), whereas this template is normally used for the synthesis of MCM-41 mesoporous materials [24,25,26]. By using CTA^+^ cation, the hydrophilic head of the amphiphile serves as an SDA for the growth of the zeolite while its hydrophobic tail generates mesoporosity, allowing the formation of hierarchical zeolite (micro/mesoporosity). It should, therefore, have superior adsorption and catalytic properties to the conventional offretite.

## 2. Results and Discussion

Initially, the crystallization of offretite was carried out at 180 °C for 40 h. The XRD results revealed that the solid product remained amorphous after 40 h, and the synthesis time was extended to 72 h. The amorphous solid completely transformed into offretite zeolite, and no other crystalline phase, such as erionite or merlinoite, was co-crystallized (Figure 1b). The XRD pattern of offretite synthesized using CTA^+^ (CTA^+^-offretite) corresponded to the simulated one, and a difference in the XRD peaks intensity of CTA^+^-offretite due to the preferred crystal orientation was observed (Figure 1a,c) [27]. The crystallization process was also performed in the absence of CTABr. However, pure LTL-type zeolite was obtained, showing the significant role of the CTA^+^ in directing the formation of offretite zeolite.

The SEM images of offretite crystals prepared using amphiphile templating route are shown in Figure 2a,b. As seen, this method successfully produced offretite single crystals with a hexagonal prismatic shape (inset of Figure 2a). The surface of CTA^+^-offretite crystals at *a*- and *b*-directions were smooth, indicating that the crystals were grown along the *c*-direction [19], forming crystals of 10.8 × 1.4 µm^2^ length with an aspect ratio of 7.7. Interestingly, the morphology of the crystals was different from that of classical TMA^+^-offretite zeolite, which exhibits a rice-grain-shape (ca. 4.6 × 3.4 µm^2^) composed of intergrown acicular primary nanocrystals (ca. 880 × 79 nm^2^) (Figure 2c,d). The smaller crystallite size of TMA^+^-offretite is the reason for the XRD peak broadening [28] (Figure 1c).

Offretite zeolite is an interesting catalyst due to its large pore size and considerably high Si/Al ratio (>3). According to the IR spectroscopy analysis, the Si/Al ratio of CTA^+^-offretite zeolite is 4.1. The molecular formula for the CTA^+^-offretite is calculated to be K_3.4_Al_3.4_Si_14_O_36_, which is very close to that of TMA^+^-offretite synthesized in the presence of KOH and NaOH mineralizers (Na_1.7_K_1.7_H_0.6_Al_4_Si_14_O_36_, Si/Al = 3.50) and the theoretically predicted one (Na_0.2_K_0.9_Al_4_Si_14_O_36_, Si/Al = 3.50) (Table 1) [5]. The textural properties of both CTA^+^-offretite and TMA^+^-offretite were studied using nitrogen adsorption–desorption analysis. The calcined CTA^+^-offretite solid exhibited a combination of type I and IV adsorption isotherms, indicating the presence of micro- and mesoporosities in the solid [29] (Figure 3a). As a result, multimodal pore size distribution was shown due to the role of CTA^+^ as a mesoporogen, as well as acting as structure-directing agent for the formation of offretite zeolite (inset of Figure 3a). The CTA^+^-offretite also had higher specific surface area (S_BET_ = 523 m^2^/g), micropore surface area (S_Mic_ = 503 m^2^/g), mesopore volume (V_Meso_ = 0.05 cm^3^/g), and total pore volume (V_tot_ = 0.26 cm^3^/g) as compared to the conventional TMA^+^-offretite zeolite (S_BET_ = 504 m^2^/g, S_Mic_ = 491 m^2^/g, V_Meso_ = 0 cm^3^/g, V_Tot_ = 0.19 cm^3^/g), which strongly supports the high crystallinity of CTA^+^-offretite zeolite (Figure 3b).

It is suggested that CTA^+^ has a dual function in the crystallization of CTA^+^-offretite. First, the trimethylamine group (R–*N*(*CH*_3_)_3_^+^, 4.26 × 2.99 Å^2^ [30]) of CTA^+^, which has nearly similar molecular size/dimensions to TMA^+^ (N(CH_3_)_4_^+^, 4.25 × 3.70 Å^2^ [30]), directs the formation of OFF framework structure via restricted electrostatic interaction with the silicoaluminate anionic oligomers (Figure 4). The inorganic intermediate species then enfold the CTA^+^ hydrophilic head, thus, inducing nucleation points for long-range propagation and crystallization of offretite zeolite. Secondly, the CTA^+^ cation also serves as a porogen by forming irregular micelles, whereby the hydrophobic tails point to each other, leading to the formation of mesopores of wide distribution [31].

The calcined CTA^+^-offretite had surface acidity after converting into protonated form (Figure 5). Hence, NH_3_-TPD analysis was used to study the acidity of the zeolite samples. As shown, both the CTA^+^- and TMA^+^-offretites had a nearly similar number of weak (T_des,150 °C_ = 0.18 mmol/g) and mild (T_des,200 °C_ = 0.25 mmol/g) acid sites. Nevertheless, the CTA^+^-offretite contained a much larger number of strong acid sites (T_des,350 °C_ = 0.32 mmol/g) than the TMA^+^-offretite (0.68 mmol/g), which could be due to its higher crystallinity and larger accessibility due to the hierarchical structure [32].

The acid properties of CTA^+^-offretite were tested in the acylation of 2-methylfuran under novel non-microwave instant heating condition (170 °C, 40 min) where the desired product 2-acetyl-5-methylfuran was obtained, which is a useful biofuel additive. The conversion was very low (12%) when no catalyst was added, indicating that acylation of 2-methylfuran is an activated reaction and, hence, requires a catalyst to overcome the activation energy [33] (Table 2, Entry 1). When hierarchical CTA^+^-offretite was added, the conversion increased tremendously to 83.5% with 100% selectivity towards 2-acetyl-5-methylfuran (molecular size 6.6 × 4.1 Å^2^ [30]) (Entry 2). The catalytic performance of CTA^+^-offretite was also compared with that of the classical TMA^+^-offretite under the same reaction conditions. As expected, when using CTA^+^-offretite, the reaction conversion (83.5%) was higher than that of the TMA^+^-counterpart (78.3%) due to its larger surface acidity and the higher accessibility of the acid sites (Entry 4). Thus, the data shows the importance of hierarchical porosity for enhanced bulk molecule diffusivity and accessibility.

The catalytic activity of CTA^+^-offretite was also compared with those of the conventional homogeneous catalysts (e.g., HCl, HNO_3_, CH_3_COOH). The results showed that CTA^+^-offretite had comparable catalytic activity to the HNO_3_ (83.6% conversion), and it exhibited even better catalytic activity than the HCl (72.3%) and the CH_3_COOH (47.1%) after 40 min of reaction (Entries 5–7). Most importantly, the CTA^+^-offretite showed high stability, and no significant loss in catalytic reactivity of the catalyst was observed, even after five consecutive runs (Entry 3). Thus, the strategy of synthesizing offretite zeolite with hierarchical structure by amphiphile-templating synthesis route can be considered as safer, cheaper, facile, and scalable compared to the previous works [34].

## 3. Materials and Methods

### 3.1. Chemicals and Materials

Potassium hydroxide pellets (KOH, 85%), ammonium nitrate (NH_4_NO_3_, 95%), nitric acid (HNO_3_, 67%), 2-methylfuran (99%), acetic anhydride (98.5%), diethyl ether (99%), and tetramethylammonium salt (TMACl) were purchased from Merck (Darmstadt, Germany). Aluminum sulfate hexadecahydrate (Al_2_(SO_4_)_3_·16H_2_O, 97%) was obtained from BDH Chemical Ltd. (Poole, England). Colloidal silica HS-40 was supplied by Sigma-Aldrich (Darmstadt, Germany). Glacial acetic acid (CH_3_COOH, 99%) and hydrochloric acid (HCl, 37%) were purchased from Qrëc (Asia) Sdn. Bhd (Rawang, Malaysia). Cetyltrimethylammonium bromide (CTABr, 97%) was procured from Acros Organics (Geel, Belgium). All chemicals were used without further purification.

### 3.2. Synthesis of CTA^+^-Offretite Zeolite

A clear aluminate solution was first prepared by dissolving KOH (3.018 g), Al_2_(SO_4_)_3_·16H_2_O (1.442 g), and CTABr (1.221 g) in distilled water (18.208 g). A clear silicate solution was then prepared by mixing HS-40 (6.875 g) with distilled water (9.958 g). A hydrogel with a final molar ratio of 20SiO_2_:1Al_2_O_3_:10K_2_O:1.46CTABr:800H_2_O was formed after mixing the silicate and aluminate solutions and subjected to aging under stirring (18 h, 25 °C, 500 rpm) and crystallization (180 °C, 72 h). The solid product was filtered, purified with distilled water until a pH of 7, and calcined at 550 °C for 6 h. Then, the obtained solid (3.000 g) was further ion-exchanged with NH_4_NO_3_ solution (1 M) and calcined (500 °C, 4 h) to produce protonated offretite zeolite. For comparison, classical TMA^+^-offretite zeolite was synthesized using tetramethylammonium salt (TMACl) as an SDA [35], and the protonated form zeolite was prepared using the same protocol.

### 3.3. Characterization

The XRD patterns were recorded with a PANalytical X’Pert PRO diffractometer (Malvern Panalytical Ltd, Malvern, United Kingdom) (Cu Kα radiation, λ = 0.15418 nm, 40 kV, 10 mA, step size = 0.02°, scan speed = 0.2 °/min, scanning range 2θ = 3°–30°). The morphological properties of zeolite samples were studied using a JEOL JSM-6701F FESEM microscope (JEOL, Tokyo, Japan) operating at 20 kV. The chemical composition (Si/Al ratio) of zeolite was estimated using the IR spectroscopy technique with a Perkin Elmer’s System 2000 spectrometer (Perkin-Elmer, Waltham, MA, USA) [36]. The porous properties of solids were studied using a Micromeritics ASAP 2010 instrument (Micrometrics, Norcross, GA, USA). Prior to measurement, the powder sample (ca. 0.07 g) was degassed at 300 °C for 12 h and the N_2_ adsorption was carried out at −196 °C. The total surface area was computed using the Brunauer–Emmett–Teller (BET) equation, whereas the external surface area and micropore volume were calculated using the t-plot model. The total pore volume of the samples was determined using the amount of N_2_ uptake at the relative pressure (p/p^o^) of 0.996. The acidity of samples was investigated with a BELCAT-B temperature programmed desorption (TPD) instrument (MicrotracBEL, Osaka, Japan). First, the powder sample (ca. 0.07 g) was outgassed at 400 °C overnight before the cell was saturated with NH_3_ gas at 25 °C for 30 min. The non-adsorbed NH_3_ was then evacuated before the sample was heated and desorbed slowly from 25 °C to 600 °C with a heating rate 10 °C/min.

### 3.4. Catalytic Test

Offretite zeolite (0.300 g, 400 °C, 4 h), 2-methylfuran (4.7 mmol), and acetic anhydride (14.1 mmol) were first added into a 10-mL glass vial. The vial was inserted into a non-microwave instant heating reactor (Monowave 50) (Anton Paar Austria GmbH, Graz, Austria) and instantly heated to 170 °C for 40 min; less than 2 min was required to reach the desired temperature. After cooling, the reaction solution was isolated using microfiltration and analyzed with an Agilent’s GCMS-5975 Turbo System and 7890A GC-FID (Santa Clara, SJ, USA).

## 4. Conclusions

In conclusion, offretite zeolite with high crystallinity has been synthesized via the amphiphile-templating approach. By using cetyltrimethylammonium bromide (CTABr) as a promising structure-directing agent (SDA), as its hydrophilic head has nearly the same dimensions as tetramethylammonium (TMA^+^), offretite zeolite crystals with hexagonal prism shape (10.8 × 1.4 µm^2^) were crystallized at 180 °C for 72 h. The resulting offretite zeolite had larger surface areas, pore volume, and pore sizes due to the use of CTA^+^, which also served as a mesoporogen. Furthermore, CTA^+^-offretite (0.74 mmol/g) had higher acidity than the conventional TMA^+^-offretite (0.68 mmol/g), and its acidity and accessible porosity were highly needed in the acylation of 2-methylfuran: 83.5% conversion and 100% selectivity to the desired product after 40 min reaction at 170 °C under non-microwave instant heating conditions were recorded. Most importantly, the CTA^+^-offretite catalyst was stable, fully recoverable, and more active than the conventional TMA^+^-offretite and other homogeneous acid catalysts, including HCl, HNO_3_, and CH_3_COOH. In short, the amphiphile-templating strategy can be applied for the preparation of other important zeolites that are usually synthesized by harmful and expensive organic templates.

## Figures and Tables

**Figure 1 molecules-26-02238-f001:**
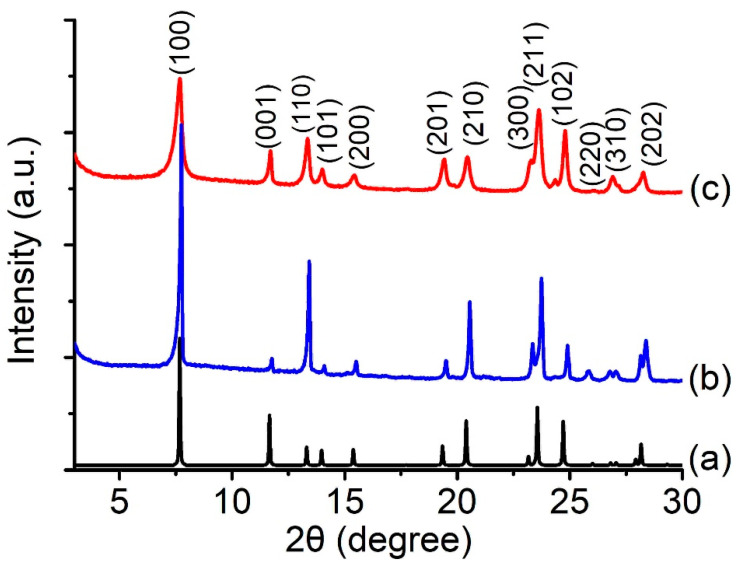
XRD patterns of (**a**) offretite (simulated pattern), (**b**) cetyltrimethylammonium (CTA^+^)-offretite and (**c**) tetramethylammonium (TMA^+^)-offretite zeolites.

**Figure 2 molecules-26-02238-f002:**
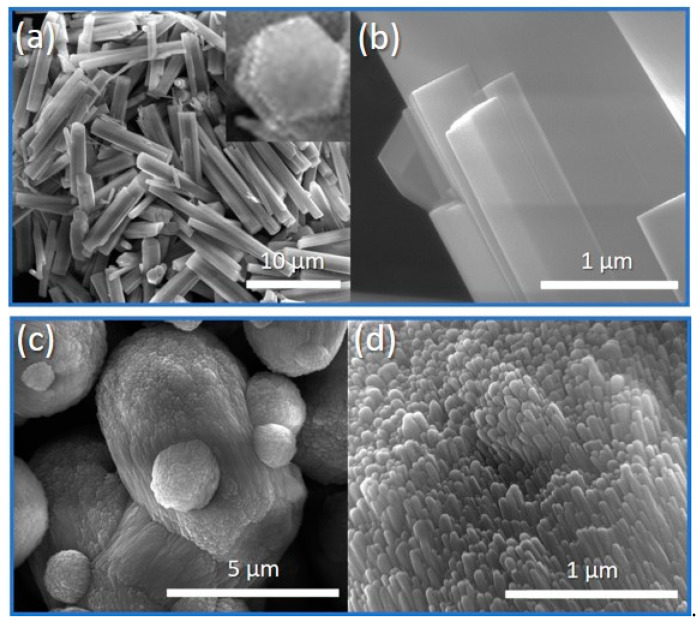
SEM images of (**a**,**b**) CTA^+^-offretite and (**c**,**d**) TMA^+^-offretite zeolites under different magnifications.

**Figure 3 molecules-26-02238-f003:**
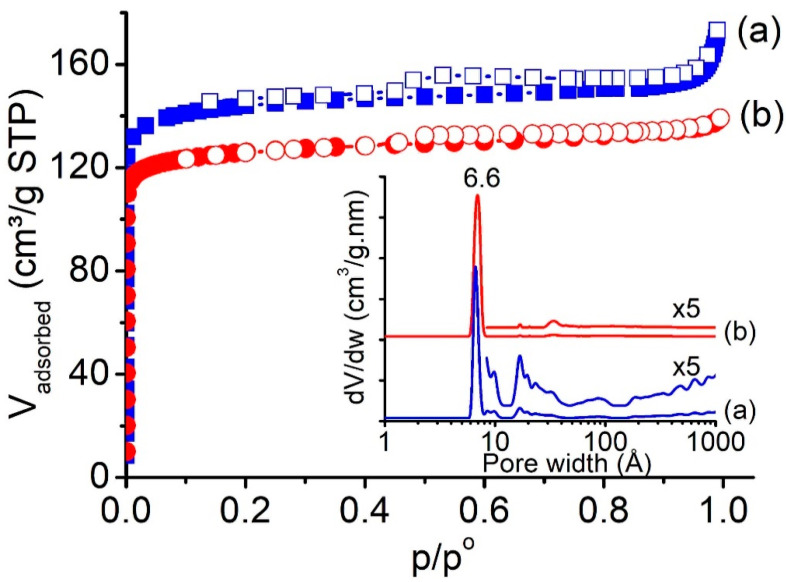
Nitrogen adsorption–desorption isotherms and (inset) pore size distributions of (**a**) CTA^+^-offretite and (**b**) TMA^+^-offretite zeolites.

**Figure 4 molecules-26-02238-f004:**
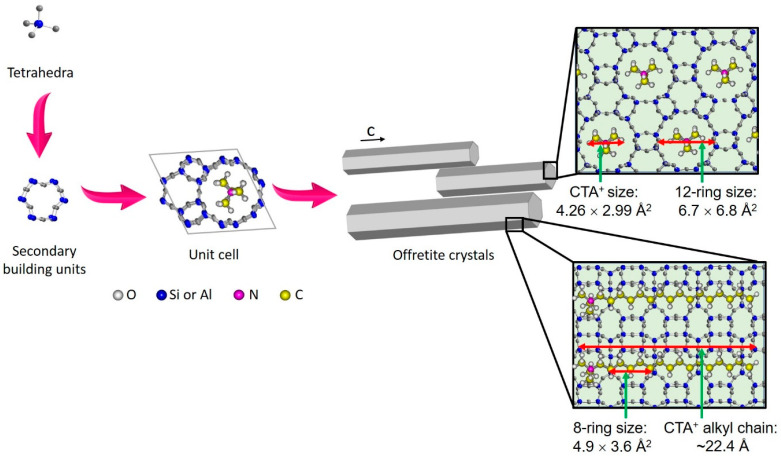
Formation mechanism of offretite zeolite templated by CTA^+^ cations (K^+^ cations are not shown for simplicity).

**Figure 5 molecules-26-02238-f005:**
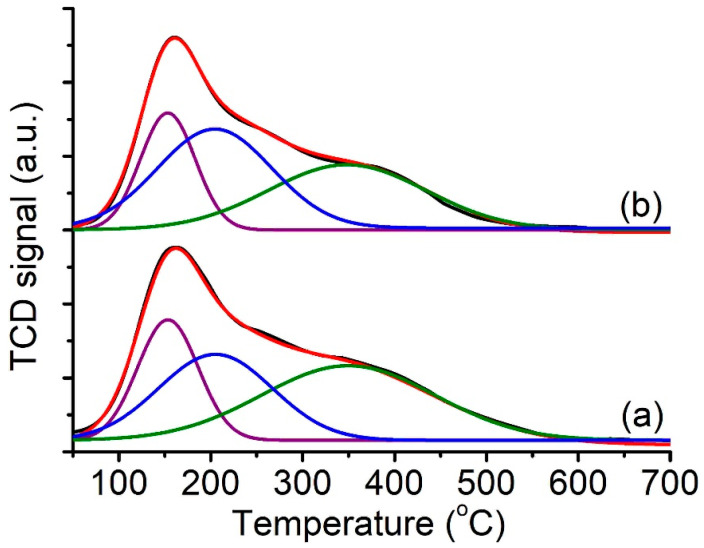
NH_3_-TPD profiles of (**a**) CTA^+^-offretite and (**b**) TMA^+^-offretite zeolites after deconvolution, which shows the presence of weak, mild, and strong acid sites.

**Table 1 molecules-26-02238-t001:** Properties of offretite zeolites.

Samples	Si/Al	S_BET_ (m^2^/g)	S_Mic_ (m^2^/g)	S_Ext_ (m^2^/g)	V_Meso_ (cm^3^/g)	V_Tot_ (cm^3^/g)	NH_3_-TPD Acidity (mmol/g)			
							T_des,_ _150 °C_	T_des,_ _200 °C_	T_des,_ _350 °C_	Total
CTA^+^-offretite	4.1	523	503	20	0.05	0.26	0.18	0.24	0.32	0.74
TMA^+^-offretite	3.5	504	491	13	0	0.19	0.17	0.26	0.25	0.68

**Table 2 molecules-26-02238-t002:** Catalytic performance of offretite catalysts in acylation of 2-methylfuran at 170 °C under non-microwave instant heating conditions.

Entry	Catalysts	Conversion (%)		
		10 min	20 min	40 min
1	No catalyst	3.2	6.4	12.0
2	CTA^+^-offretite	52.6	72.3	83.5
3	CTA^+^-offretite ^a^	50.2	68.0	79.1
4	TMA^+^-offretite	44.7	61.7	78.3
5	HCl	35.7	55.3	72.3
6	HNO_3_	51.1	71.9	83.6
7	CH_3_COOH	16.0	31.9	47.1

^a^ after fifth run.

## Data Availability

Not applicable.

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
