# Peer review of "Offretite Zeolite Single Crystals Synthesized by Amphiphile-Templating Approach"

_molecules, 2021, doi:10.3390/molecules26082238_

Round 1

Reviewer 1 Report

The manuscript “Offretite Zeolite Single Crystals Synthesized by Amphiphile-2 Templating Approach” describes different synthesis route for OFF zeolite using CTAB. The research is interesting and well-presented, explaining some mechanistic differences from the synthesis that employs TMA+ cations. Still, there are two minor aspects to clear:

  1. The authors claimed that CTA+ induced mesoporosity in the sample. This fact is according to the isotherms showing a type H4 hysteresis (Thommes et al., 2015). I suggest adding this information and use the isotherms to calculate the mesoporous volume, as only the total pore volume is shown.
  2. In Materials and Methods, the calcination temperature process differed from the synthesis and the catalytic tests (lines 189 and 211). I suppose it is a typo. If not, were there any differences between the materials calcined at those temperatures? Was there any remaining OSDA when calcined at the lower temperature?

Author Response

The manuscript “Offretite Zeolite Single Crystals Synthesized by Amphiphile-2 Templating Approach” describes different synthesis route for OFF zeolite using CTAB. The research is interesting and well-presented, explaining some mechanistic differences from the synthesis that employs TMA+ cations.

The authors would like to thank Reviewer #1 for his/her constructive comments.

Still, there are two minor aspects to clear:

1. The authors claimed that CTA+ induced mesoporosity in the sample. This fact is according to the isotherms showing a type H4 hysteresis (Thommes et al., 2015). I suggest adding this information and use the isotherms to calculate the mesoporous volume, as only the total pore volume is shown.

The mesopore volumes (VMeso) have been calculated and added into the manuscript. Please see page 6.

2. In Materials and Methods, the calcination temperature process differed from the synthesis and the catalytic tests (lines 189 and 211). I suppose it is a typo. If not, were there any differences between the materials calcined at those temperatures? Was there any remaining OSDA when calcined at the lower temperature?

The calcination temperature (550 °C) is correct. 550 °C is needed to completely remove the occluded OSDA. The OSDA cannot be remove if the calcination temperature of 180 °C (synthesis temperature) or 170 °C (catalytic reaction temperature) is used.

Reviewer 2 Report

In this work, Ng and coauthors present a novel approach to synthesize pristine crystals of offretite zeolite, which improves the method currently used to synthesize it. Moreover, they performed an interesting study of its catalytical properties, including porosity and acid sites. The paper is well-written and the methodology well-designed. The results, discussion and conclusions are clear and easy to follow. For all these reasons, I recommend its publication after considering the following suggestions. Suggestions are left to the author's discretion whether to follow them or not.

Suggestion:

Figure 1 will be easy to identify if each diffractogram has a different colour. In addition, it must be considered that the journal is electronic (i.e. there is no additional fees to publish in colour).

I recommend improving the caption of figure 5. The public not specialised in TPD method will want to know the meaning of the different colours used in the graph.

Author Response

In this work, Ng and coauthors present a novel approach to synthesize pristine crystals of offretite zeolite, which improves the method currently used to synthesize it. Moreover, they performed an interesting study of its catalytical properties, including porosity and acid sites. The paper is well-written and the methodology well-designed. The results, discussion and conclusions are clear and easy to follow. For all these reasons, I recommend its publication after considering the following suggestions. Suggestions are left to the author's discretion whether to follow them or not.

 The authors would like to thank Reviewer #2 for his/her constructive comments.

Suggestion:

Figure 1 will be easy to identify if each diffractogram has a different colour. In addition, it must be considered that the journal is electronic (i.e. there is no additional fees to publish in colour).

The data in Figures 1 and 3 has been presented in colour for ease identification.

I recommend improving the caption of figure 5. The public not specialised in TPD method will want to know the meaning of the different colours used in the graph.

The caption of Figure 5 has been revised by adding more detailed information about TPD peak deconvolution.